# Liver Ultrasound Histotripsy: Novel Analysis of the Histotripsy Site Cell Constituents with Implications for Histotripsy Application in Cell Transplantation and Cancer Therapy

**DOI:** 10.3390/bioengineering10020276

**Published:** 2023-02-20

**Authors:** Saied Froghi, Matheus Oliveira de Andrade, Layla Mohammad Hadi, Pierre Gelat, Hassan Rashidi, Alberto Quaglia, Barry Fuller, Nader Saffari, Brian Davidson

**Affiliations:** 1Department of HPB & Liver Transplantation Surgery, Royal Free London NHS Foundation Trust, Pond Street, Hampstead, London NW3 2QG, UK; 2Centre for Surgical Innovation, Organ Regeneration and Transplantation, UCL Division of Surgery & Interventional Sciences, Royal Free Hospital Campus, Pond Street, Hampstead, London NW3 2QG, UK; 3Ultrasonics Group, Department of Mechanical Engineering, Roberts Engineering Building, University College London, Torrington Place, London WC1E 7JE, UK; 4Stem Cell & Regenerative Medicine Section, UCL Great Ormond Street Institute of Child Health, London WC1N 1EH, UK; 5Department of Cellular Pathology, Royal Free London NHS Foundation Trust, Pond Street, Hampstead, London NW3 2QG, UK

**Keywords:** HIFU, high-intensity focused ultrasound, boiling histotripsy, liver transplantation, cell transplantation, adult hepatocyte, cell isolation

## Abstract

**Introduction:** Allogenic hepatocyte transplantation is an attractive alternative to whole-organ transplantation, particularly for the treatment of metabolic disorders and acute liver failure. However, the shortage of human donor organs for cell isolation, the low cell yield from decellularisation regimes, and low engraftment rates from portal administration of donor cells have restricted its clinical application. Using ultrasound histotripsy to provide a nidus in the liver for direct cell transplantation offers a new approach to overcoming key limitations in current cell therapy. We have analysed the liver cavity constituents to assess their potential as a site for cell delivery and implantation. **Methods:** Using human organ retrieval techniques, pig livers were collected from the abattoir and transported in ice-cold storage to the laboratory. Following 2 h of cold storage, the livers were flushed with organ preservation solution and placed on an organ perfusion circuit to maintain viability. Organs were perfused with Soltran™ organ preservation solution via the portal vein at a temperature of 24–30 °C. The perfusion circuit was oxygenated through equilibration with room air. Perfused livers (n=5) were subjected to ultrasound histotripsy, producing a total of 130 lesions. Lesions were generated by applying 50 pulses at 1 Hz pulse repetition frequency and 1% duty cycle using a single element 2 MHz bowl-shaped transducer (Sonic Concepts, H-148). Following histotripsy, a focal liver lesion was produced, which had a liquid centre. The fluid from each lesion was aspirated and cultured in medium (RPMI) at 37 °C in an incubator. Cell cultures were analysed at 1 and 7 days for cell viability and a live-dead assay was performed. The histotripsy sites were excised following aspiration and H&E staining was used to characterise the liver lesions. Cell morphology was determined by histology. **Results:** Histotripsy created a subcapsular lesion (~5 mm below the liver capsule; size ranging from 3 to 5 mm), which contained a suspension of cells. On average, 61×10^4^ cells per mL were isolated. Hepatocytes were present in the aspirate, were viable at 24 h post isolation and remained viable in culture for up to 1 week, as determined by phalloidin/DAPI cell viability stains. Cultures up to 21 days revealed metabolically active live hepatocyte. Live-dead assays confirmed hepatocyte viability at 1 week (Day 1: 12% to Day 7: 45% live cells; *p* < 0.0001), which retained metabolic activity and morphology, confirmed on assay and microscopy. Cell Titre-Glo^TM^ showed a peak metabolic activity at 1 week (average luminescence 24.6 RLU; *p* < 0.0001) post-culture compared with the control (culture medium alone), reduced to 1/3 of peak level (7.85 RLU) by day 21. **Conclusions:** Histotripsy of the liver allows isolation and culture of hepatocytes with a high rate of viability after 1 week in culture. Reproducing these findings using human livers may lead to wide clinical applications in cell therapy.

## 1. Introduction

Liver transplantation is the only curative treatment for patients with end-stage liver disease and liver-based metabolic disorders. Such an intervention is limited by the shortage of donor organs [1,2]. A potential alternative to liver transplantation is allogenic hepatocyte transplantation. Over the last two decades, hepatocyte transplantation has made the transition from bench to bedside [1,2]. Standardised techniques have been established for the isolation, culture, and cryopreservation of human hepatocytes, which has led to the expansion of clinical programmes [3]. Clinical hepatocyte transplantation safety and short-term efficacy have been demonstrated [4,5,6], with limited clinical impact. A major challenge hampering the wider clinical adoption of this technique is the shortage of donor organs from which to isolate hepatocytes, as viable organs are prioritised for whole-organ transplants. Only sub-optimal grafts that have been declined for whole-organ transplants are currently being offered for cell isolation. This may partly explain the low cell yields of hepatocyte transplants. However, current methods for engraftment are also highly inefficient, resulting in a limited clinical application [7]. In addition to the above challenges, a major risk to the recipient from portal administration of cells is portal vein thrombosis [8].

Current hepatocyte isolation techniques usually involve a combination of mechanical disruption of the liver tissue and the perfusion of collagenase via the major hepatic veins [7,9,10]. The mechanical and chemical treatment damages the harvested cells and results in low cell yields [10]. In addition, the current success of cell engraftment is dependent on the number of viable cells extracted and subsequently implanted. Repeated hepatocyte transplantation has been shown to increase the number of engrafted cells above 5% of recipient liver cell mass, a level sufficient to correct some metabolic defects [11]. Primary adult hepatocytes lose function and viability following isolation and culture, and have limited proliferation potential in vitro [11].

A new and radically different approach is clearly required for both isolation and engraftment to improve the outcome of cell therapies for both acute and chronic liver disease.

High-intensity focused ultrasound (HIFU) has been used to treat cancers through thermal ablation [12,13]. A new type of HIFU therapy—boiling histotripsy (BH)—can mechanically dissociate tissue with a range of accompanying mechanical and thermal effects that can be tuned depending on the requirements of the application [13,14].

Histotripsy has also been investigated as a method for treating cancers. In BH, localised, high-amplitude shock waves cause rapid tissue heating, resulting in the transient formation of a boiling bubble within the tissue [15]. The interaction of incident ultrasound (US) shocks with the boiling bubble results in tissue disruption and liquefaction without significant thermal injury to the tissue. This happens because the timescales of heat diffusion and thermal injury are much longer than those of the mechanical action of oscillating bubbles [14,16,17].

This property of histotripsy led to experimental work on the decellularisation of tissue whilst maintaining a degree of extracellular structure (i.e., preserved blood vessels at the site of HIFU insonation) [12,18,19]. Histotripsy has not previously been applied to hepatocyte isolation from liver tissue or as a potential technique to facilitate cell transplantation. Moreover, the vast majority of current cell-extraction techniques [2,3,4,5,6,7,9] have been based on the collagenase digestion technique, originally developed by Berry & Friend [20], which is both time consuming and expensive. In this paper, we investigate the cell components of the liver histotripsy cavity as an important aspect of a potential niche for cell implantation. Perfusion of the isolated liver maintains tissue viability and vessel patency, and has been shown to influence the quality of the lesion created by the HIFU system [21].

## 2. Materials and Methods

To investigate the possible use of histotripsy in ex vivo perfused porcine livers for their suitability as a nidus for cell transplantation as a novel technique and assessing lesions histologically and its constituents for viability.

### 2.1. Overall Experimental Method

Porcine livers were retrieved fresh from the abattoir. They were obtained within 10 min of the termination (warm ischaemia time) at the slaughterhouse. Following termination, the livers were rapidly retrieved using human organ retrieval techniques. These consisted of en-bloc retrieval of the pig’s abdominal viscera followed by on-table isolation of the liver, with the major vessels and main bile duct. A subsequent on-table vessel perfusion was used to clear to remaining blood from the liver. This was achieved by perfusing the liver grafts with 1000 mL of heparin saline solution via the portal vein and transferring them to the organ perfusion laboratory in an organ preservation solution (saline was used as preservation solution for transport), on packed ice at about 5 °C, in an insulated organ storage box (average cold storage time: 2 h). On arrival at the organ perfusion laboratory, the liver grafts were transferred to an organ perfusion circuit, which maintained organ viability by perfusion with Soltran organ preservation solution (Baxter Healthcare, Newbury, UK) using the protocol described in Figure 1.

Once established on the perfusion circuit, each liver was subjected to a histotripsy protocol. The site of histotripsy was selected at random, with both peripheral and central portions of the liver included for histological analysis and cell culture. Immediately following histotripsy treatment, lesion contents were aspirated from the core of each histotripsy lesion and were cultured in a 96-well plate. Aspirates were examined for cell content, quantity and viability of cells, and were then cultured (Figure 1). Following culture, the cells were examined under light microscopy for morphology and subsequently fixed for live-dead assay and alamarBlue^TM^ stains (Thermo Fischer Scientific, Hemel Hempstead, UK). The experiments were repeated with five different porcine livers using the same histotripsy parameters and protocol.

#### 2.1.1. HIFU Set Up

A 2 MHz single element bowl shaped US transducer (Sonic Concepts H-148, Bothell, WA, USA) with an aperture size of 64 mm and a 22.6 mm central aperture was used with a transparent coupling cone (Sonic Concepts, C-101, Bothell, WA, USA) filled with degassed, de-ionised water. This transducer has an axial focal zone length of 5.72 mm with a lateral extent of 0.76 mm. With a focal length of 63.2 mm for this transducer, the centre of the lesion would be at a depth of 5.6 mm below the surface of the tissue sample. The transducer was driven by two function generators (Agilent 33220A, Santa Clara, CA, USA) in series via a linear radiofrequency (RF) power amplifier (ENI 1040 L, Rochester, NY, USA). The first function generator was set to generate 50 cycles of a 1 Hz rectangular wave with 1% duty cycle. This triggered the second function generator, which output a 2 MHz sinusoidal wave into the RF power amplifier, resulting in 50 pulses each with a 10 ms duration. A power meter (Sonic Concepts 22A, Bothell, WA, USA) was connected between the RF amplifier and the ultrasound transducer, and the electrical power supplied to the transducer was monitored to be approximately 150 W. The pulse-averaged electrical power was 1.5 W (calculated using *P*_average_ = *P*_peak_ × duty cycle). During the experiments, an acoustic absorber (AptFlex F28, Precision Acoustics Ltd., Dorchester, UK) was placed under the liver samples to minimise acoustic reflection.

#### 2.1.2. Modelling to Define the Histotripsy Parameters

HIFU parameter modelling:

A modelling approach was used to estimate the HIFU pressure fields at the treatment zone. The HITU Simulator v2.0 [22,23] was used to solve the Khokhlov–Zabolotskaya–Kuznetsov (KZK) equation and obtain 2D axisymmetric pressure waveforms along the propagation axis and the radial direction. Both the KZK equation and the HITU Simulator have been extensively used to model boiling histotripsy pressure and temperature fields [15,17,23,24].

The acoustic simulations were performed for a single-element bowl-shaped transducer operating at 2 MHz (Sonic Concepts H-148) for input electrical powers of 150 W assuming 85% transducer efficiency. Acoustic propagation was numerically evaluated in a domain consisting of 5.76 cm of water along the axial direction, followed by a semi-infinite region of liver tissue. To simulate the experimental set-up for histotripsy in perfused livers described in the sections below, acoustic field quantities were evaluated at 0.56 cm into the liver domain. The spatial grid for acoustic consisted of 10 elements per wavelength in the axial direction and 15 elements per wavelength in the radial direction. Peak focal pressures obtained were $P^{+}=76.7$ MPa and $P^{-}=13.4$ MPa, with focal heating rates of 935 W·cm^−3^ and a maximum acoustic intensity of 20.0 kW·cm^−2^.

#### 2.1.3. Organ Perfusion, Histotripsy Lesion Formation and Assessment Perfusion Set Up

Upon arrival at the organ perfusion laboratory, the livers were placed in an organ bath lying over an ultrasound reflective layer and perfused with 1 L of preservation solution (Soltran, Baxter, Newbury, UK), and the core liver temperature was allowed to return to room temperature (ranging between 24 and 30 °C) before being subjected to histotripsy insonation^1^. Perfusion of the organ to maintain viability was administered via the portal vein (Figure 2) with the perfusate draining via the vena cava into an organ bath. The perfusate was not recycled. The perfusion solution was delivered using a perfusion pump (Baxter™, Newbury, UK) to achieve a constant flow rate of 350 mL·h^−1^, confirming vessel patency before lesions were created by US histotripsy. The HIFU probe was positioned in multiple sequential locations chosen at random over the surface of the perfused liver with 50 pulses applied over approximately one minute at each site. The focal histotripsy lesions could be identified by the puckering of the liver capsule, a pinpoint dimple. Once the lesions were created, these were incised with a surgical blade and the central liquefied core was aspirated using a 20 µL single-channel gauge pipette. The aspirate was subject to pre-culture microscopy and then immediately transferred into culture medium for subsequent assessment of cell number and morphology.

### 2.2. Pre-Culture Light Microscopy

Prior to culture, the aspirates were examined under light microscopy and confirmed to have a mixture of cells and debris. As the aspirate underwent a spin process prior to culture, they were not stained at this stage to identify the cell type.

### 2.3. Cell Culture: Cell Morphology and Growth

Following aspiration of the histotripsy lesions, the samples were placed immediately in cryovials to which 150 μL of RPMI cell culture media (Gibco, Thermo Fisher Scientific, Hemel Hempstead, UK) was added, supplemented with 10% FBS (Thermo Fisher Scientific, Hemel Hempstead, UK), 1% penicillin (5000 units/mL) and streptomycin (5000 μg·mL^−1^) (Thermo Fisher Scientific, Hemel Hempstead, UK). Each sample was strained (40 µm filter) to remove debris (VWR Collection, VWR, Leicester, UK) and the cells within the sample were cultured in a 24-well plate (Corning Costar, Sima Aldrich, Dorset, UK) for up to 21 days. The cells were stained at different time points using a live-dead assay and their growth was monitored using phalloidin/DAPI staining. Three aspirates were cultured in each well. Each lesion was aspirated once and then the lesion was excised for histological assessment. Before culture, a light microscopy assessment was carried out after suspension of the cell aspirate to ensure each well contained cells rather than debris.

### 2.4. Phalloidin/DAPI Staining

Cell morphology and growth at days 3 and 7 post seeding were examined using fluorescently labelled phalloidin, which allowed filamentous actin to be visualised, and DAPI, which allowed the cell nuclei to be observed. Following the manufacturer’s instructions (Abcam, Cambridge, UK), the cells were fixed using formalin (10%) for 15 min and then washed with PBS. They then underwent permeabilisation with 1% BSA and 0.3% Triton-X solution for 30 min before being stained with phalloidin (2.5% in BSA) Triton-X solution for 90 min. The cells were then washed with PBS (3×) and stained with DAPI for 10 min before imaging. The cells were imaged using an EVOS fluorescence inverted microscope (EVOS FL colour, Life Technologies, Carlsbad, CA, UK) at wavelengths (Ex/Em 495/518 nm) for phalloidin and (Ex/Em 352/461 nm) for DAPI. Phalloidin functions by binding and stabilising filamentous actin (F-actin) and effectively prevents the depolymerisation of actin fibres. It is used with a fluorescent tag to reveal the cell cytoskeleton. DAPI is a fluorescent stain that binds strongly to adenine–thymine rich regions in DNA and identifies active dividing cells. DAPI staining has 94% overall sensitivity for cell cycle profiling and therefore allows an integrative and simultaneous quantitative analysis of molecular and morphological parameters [25].

### 2.5. Live-Dead Assay

The viability of the cells was assessed on days 3, 7 and 21 using a live-dead imaging kit (Molecular Probes, Thermo Fisher Scientific, Hemel Hempstead, UK). As per the manufacturer’s guidelines, the cells underwent incubation with live-dead solution containing 0.05% of 4 mM Cacein- AM (Ex/Em: 495/515 nm) and 0.2% of 2 mM Ethidium homodimer-1 (Ex/Em 495/635 nm) at room temperature for 30 min prior to imaging them with an EVOS fluorescence inverted microscope (EVOS FL color, Life Technologies, Carlsbad, CA, US). The Live/Dead Cell Double Staining Kit is utilised for simultaneous fluorescence staining of viable and dead cells. This kit contains calcein-AM and ethidium solutions, which stain viable and dead cells, respectively. Calcein-AM, an acetoxymethyl ester of calcein, is highly lipophilic and cell membrane permeable. Though calcein-AM itself is not a fluorescent molecule, the calcein generated from calcein-AM by esterase in a viable cell emits a strong green fluorescence (λex 490 nm, λem 515 nm). Therefore, calcein-AM only stains viable cells. Alternatively, the nuclei staining dye ethidium cannot pass through a viable cell membrane. It reaches the nucleus by passing through disordered areas of dead cell membrane and intercalates with the DNA double helix of the cell to emit red fluorescence (λex 535 nm, λem 617 nm). Since both calcein and ethidium-DNA can be excited with 490 nm light, simultaneous monitoring of viable and dead cells is possible with a fluorescence microscope. The percentage of live and dead cells was calculated after staining.

### 2.6. Morphology Assessment

Once stained, the cells were examined under the microscope for morphology and confirmed as hepatocytes following review by two independent researchers and subsequently by an experienced hepatopathologist.

### 2.7. Cell Titre-Glo Metabolic Assay

Cell Titre-Glo (CTG) 3D cell (Promega, Southampton, UK) viability assay was used to measure ATP levels of the isolated hepatocytes in each well. The presence of metabolically active cells was quantified using the luminescence given off by the ATP produced. This assay was found to be appropriate since cells extracted from tissue have the tendency to generate ECM when cultured. This type of bioluminescence assay is found to be more sensitive in low-density cell populations [26]. In addition, this assay is known to have a high sensitivity for cell proliferation and cell toxicity. Given the low volume of cells we were working with, this assay proved to be ideal in assessing the metabolic activity of live cells. The cultures and CTG reagent were equilibrated to room temperature for 15 min before use. Cell-culture supernatant was removed from each well to leave 50 µL in each well prior to adding the reagent. A total of 50 µL of CTG reagent was then added to each well and the plate was placed on a shaker at 900 rpm for 30 s. The plate was then covered with foil and incubated at room temperature for 30 min. Finally, the luminescence was measured using a plate reader (TECAN, Infinite^®^ M200 PRO, Reading, UK). One well containing only culture medium and no cells was used as control. Luminescence from each well containing cells was measured on days 7 and 21 post culture.

### 2.8. Histological Evaluation of the Excised Histotripsy Sites

Haematoxylin & eosin (H&E) staining of the excised histotripsy sites was used to identify important structural information and any change following sonication to the liver parenchyma.

Picrosirius red stain was used to demonstrate structural and architectural damage to collagen following histotripsy by staining for collagen type I & III. A single 4 µm paraffin section from each sample was stained with haematoxylin (Harris Haematoxylin, Shandon, UK) and eosin (Eosin Y, Shandon, UK) (H&E) using an auto-stainer (Leica, ST5020, Milton Keynes, UK).

One additional section from each sample was stained with Picrosirius red (SR) stain (Sigma-Aldrich, Direct Red 80, Dorset, UK) with the following protocol: (1) deparaffinise sections in two changes of xylene. (2) Rehydrate through two changes of alcohol, 70% alcohol and distilled water. (3) Apply SR solution to cover entire sections and incubate in humidified chamber for 60 min. (4) After 60 min, rinse slides quickly in two changes of acetic acid solution. (5) Rinse slides in absolute alcohol and then dehydrate through two changes of alcohol, clear in two changes of xylene and mount with pertex mounting medium.

A further section from each sample was stained for reticulin. The staining procedure was carried out using the Reticulum II stainer kit (Roche catalogue number 860–024) and a Ventana BenchMark Special Stain system.

### 2.9. ImageJ Software

ImageJ 1.52 (National Institute of Health, Bethesda, MD, USA) was used in processing and optimisation of the images acquired from histology and cell microscopy. The Cell Count plugin was used for counting cells in the live-dead and DAPI microscopy images. The Figure J plugin was used in construction of optimised images for publication. https://imagej.nih.gov/ij/index.html (Access date: 20 June 2020)

### 2.10. Statistical Methods

GraphPad Prism^©^ 6 software was used in the analysis of the data. The Chi-squared test was used to compare the proportion of live cells on day 1 and day 7 post culture. The Student’s *t*-test or Mann–Whitney test was used to determine the difference, which was set at *p* < 0.05. We did not use statistical methods to predetermine sample size, there was no randomisation designed in the experiments, and the studies were not blinded. Data are represented as mean ±SEM or median where appropriate.

## 3. Results

### 3.1. Organ Perfusion and Viability, Histotripsy Lesion Creation and Core Aspiration

All livers were successfully flushed and achieved uniform perfusion based on the visual appearance of the organ. We analysed 130 individual lesions from five livers using the protocol described above (Figure 1). During perfusion, all livers maintained a degree of bile production, suggesting they were viable livers capable of active bile production and excretion. As described previously, following treatment with histotripsy, each lesion was bisected and the fluid contents at the core of the lesion were then aspirated. The time taken to aspirate each lesion was less than a minute. To increase the volume of cell suspension, the aspirate from three sequential lesions was combined in each well of the 96-well culture plate.

### 3.2. Histology of Lesions

The histotripsy lesions were evaluated by an experienced liver histopathologist (AQ). The lesions were typically about 0.5 cm below the surface of the liver capsule and the central core was liquified (Figure 3). The liquified core, which had been aspirated, contained a suspension of cells mixed with extracellular matrix debris (Figure 3 and Figure 4). Cavity size ranged from 3 to 5 mm. Accurate measurement of the length of the lesion was difficult in some instances due to the orientation of the lesion once fixed. The typical HIFU histotripsy site appeared histologically to track from the subcapsular region to an intraparenchymal depth of about 0.5–1 cm. Except for sampling and embedding artefacts, the liver capsule was intact. The histotripsy site consisted of an area of loss of hepatic plates and supporting matrix, with rupture of intervening interlobular septa and a few instances of portal structures. In two cases, the ruptured portal, biliary and vascular structures were small (artery and bile duct of 125 μm diameter, and portal vein maximum diameter 250 μm) but in one case the histotripsy broke the wall of a larger portal vein branch of 800 µm diameter. The histotripsy area was sharply demarcated with hepatocytes a few micrometres away from its edge, appearing intact morphologically. There was no difference in the histological appearance of the lesions from the periphery of the liver compared with more central lesions.

### 3.3. Cell Culture

Immediately after the cells were centrifuged and filtered, they were cultured in 96-well plates with RPMI and antibiotics. Wells remained free of infection, and none were discarded.

### 3.4. Cell Type, Appearances, and Numbers at Baseline

An initial manual count of the cells within the aspirate revealed approximately 2446 cells/aspirate, equating to 61 × 10^4^ cells per mL. Although the number of cells aspirated from each lesion was variable (Figure 4), the count was approximately 37 × 10^4^ cells in 600 µL of aspirate. Given each well contained three aspirates, the total number of cells cultured per well was approximately 1.15 × 10^6^. Initial cell aspirates contained cell debris along with a mixture of different cells (hepatocytes, endothelial cells and fibroblasts; identified based on microscopic morphological appearance by an experienced hepato-pathologist (AQ)). Extracellular matrix components were also present (Figure 4). A live-dead staining analysis along with manual count of cells on day 1 revealed that 12–16% of the cell population was viable (approximately n=1523 of 7682 cells per well) (Figure 5).

### 3.5. Cell Division and Viability in Culture

DAPI staining of cells 7 days post culture revealed live cells that had started to attach to the culture plates, spread and associate with neighbouring cells (Figure 6). The cells had retained morphology and had an intact cytoskeleton, and had started to form a network. The average number of live cells per well almost doubled between baseline and day 7 (day 1: 1206 cells to day 7: 2022 per well) (Figure 7) and there was a change from 12% to 45% live cells after 7 days of culture. The increase in viable cell numbers was statistically significant (*p* < 0.0001) (Table 1). Morphological analysis of the live cells on the 7th day confirmed adult hepatocyte replication. Metabolic activity of the cultured cells analysed using a cell titre-Glo assay showed an activity peak 7 days post culture (Figure 8) and further confirmed survival of cells. This was a significant increase when compared with control luminescence (medium alone) (*p* < 0.0001); luminescence: 3.57 RLU).

Although there was a decline in average metabolic activity per well from the first week onwards (average luminescence: 24.6 RLU), cultured cells also displayed metabolic activity 21 days post culture (average luminescence: 7.85 RLU) when compared with the control group. Metabolic activity was demonstrated across all wells over the 21 days of culture monitoring.

## 4. Discussion

We report a detailed analysis of the local histological changes following liver histotripsy of viable whole liver organs in vitro using an isolated perfused organ system. A unique observation was a consistent hepatocyte isolation and expansion in culture following aspiration of the histotripsy cavity content, which has not been previously described. The use of histotripsy to extract cells in the perfused whole-organ system can provide an alternative to collagenase cell extraction. Histotripsy can help not only in cell extraction but also in implantation of extracted cells by creating a niche environment for the cells.

### 4.1. Treatment Dose and the Nature of the Liver Lesion Based on Histology

There are several factors that affect the extent and nature of the histotripsy lesion in the liver. The histotripsy treatment dose applied is important in creating a lesion in which the core consists of a liquified suspension.

Khokhlova et al. [13] noted that the histotripsy area is sharply demarcated and hepatocytes a few micrometres away from the lesion edge appear morphologically intact. We have noted similar findings in our study. However, with our treatment protocol we have seen damage to branches of a portal vein larger than those observed by Khokhlova et al. [13].

This finding may have important clinical implications as it highlights the importance of appropriate dose adjustment for the tissue being sonicated [27]. For example, the dose applied for producing a protective niche for cell implantation would be different from that applied for tumour ablation. A break in vasculature in a clinical setting could be of concern in terms of bleeding if being used for focal cavity formation and could allow dissemination of tumour cells, when using histotripsy to treat neoplastic lesions. Worlikar et al. [28] demonstrated tumour regrowth following treatment of murine HCC with histotripsy. Similar findings were reported by Ruger et al. [29] in canine bone tumours post treatment. Our study, which has revealed for the first time the presence of live cells within the core lesion post histotripsy, could explain the finding of tumour regrowth if histotripsy for cancers similarly resulted in a liquid core of viable cancer cells. Although the focus of this research programme was liver regeneration and not cancer treatment, this finding may be highly relevant to tumour treatment using histotripsy.

The organ perfusion system was used to maintain organ viability, but Soltran organ preservation solution might introduce stabilised gas microbubbles in the vasculature, which are likely to oscillate as a response to the incoming acoustic field [30]. These pre-existing bubbles could then cause mechanical damage to the vascular wall to a higher extent than bubbles that nucleate “from scratch”, i.e., solely as a response to the acoustic field, like those observed in intrinsic threshold histotripsy and boiling histotripsy protocols [13].

With intrinsic threshold histotripsy, damage to the endothelium inside the veins in proximity to the lesion and complete coagulation necrosis of the vascular wall were reported by Vlaisavljevich et al. [31].

The HIFU setup used in these experiments was based on previous experience of the group with histotripsy protocols for the focal mechanical disintegration of bovine, porcine and murine liver tissue. In our previous work [16], we monitored acoustic emissions during boiling histotripsy using a Passive Cavitation Detector (PCD). The PCD showed significant broadband emissions due to inertial cavitation activity. In fact, the point in time at which a boiling bubble was created and the subsequent excitation of a cavitation cloud could be seen in the analysis of PCD signal spectrograms.

In view of this previous work, we expected to have excited sufficient bubble activity for complete tissue fractionation. Fifty HIFU pulses were used to achieve mechanical disintegration of the treatment area. Previous works have used 5 to 50 pulses and report complete focal tissue emulsification (while surrounding tissue is preserved) upon further histological analysis [19,32]. Previous experiments in tissue phantoms show that the degree of mechanical fractionation of the treatment zone increases with increasing pulses [17]. None of the previous studies have analysed content of the histotripsy lesions.

In our study, a Pulse repetition frequency (PRF) of 1 Hz was used so that the focal region had time to cool down before subsequent pulses to avoid thermal damage. Following the simulation studies, a 1% duty cycle was used in order to raise the focal temperature so that focal peak-negative pressures surpassed the temperature-dependent nucleation threshold of soft tissue [15] for sonication at an ultrasound frequency of 2 MHz. By optimising the duty cycle of our experiments in terms of the minimal pulse lengths required for bubble nucleation, it was likely that thermal damage inflicted by HIFU heat deposition was minimised, contributing to increased viability of cells within the treatment zone. These results highlight the importance of HIFU parameters in relation to the biophysical effect and in achieving the desired outcomes (i.e., cavity formation vs. tissue destruction). A future formal comparison of histotripsy protocols would substantiate these results and help further with parameterisation for different therapeutic purposes.

### 4.2. The Use of the Viable Perfused Organ as a Model of In Vivo Histotripsy

Perfusion maintains viability of the liver ex vivo to allow intervention and evaluation similar to the circumstances in vivo [33]. Thus, perfusion may play an important part in the presence of viable cells in the histotripsy cavity isolate, as perfused organ setup is more likely to reflect tissue physiology in vivo. In our study, perfused livers produced a lesion with a core suspension that was aspirated and cultured successfully. This was repeatable from many lesions across different surface sites of the liver. The liver is a highly vascular organ, and the circulation creates a heat-sink effect, disseminating the heat from an area of thermal injury. However, one should bear in mind that the timescales of histotripsy are of the order of milliseconds, whilst the timescales of perfusion and heat dissemination are of the order of seconds. Hence, other factors in the perfused model could influence the cell isolation. Since the histotripsy mechanism is non-thermal, the heat-sink effect from liver blood vessels adjacent to the histotripsy site is unlikely to influence the size or nature of the histotripsy cavity and its contents [34]. The cavitation cloud initiation threshold changes with temperature as well as the tissue’s tensile strength, density and water content [35,36]. The water content of the tissue in the perfused model will be higher than that of a non-perfused liver. Vlaisavljevich et al. [36] demonstrated in 43 types of harvested porcine tissue the importance of tissue density, water content, ultimate stress and ultimate fractional strain in determining tissue erosion after sonication. The nature of the tissue affects the histotripsy profile and rates of heat deposition/transfer around the lesion and makes parametrisation tailored to the specific tissue an important aspect to consider [35]. The perfused organ system may have been key to the isolation of viable hepatocytes from the histotripsy cavity just as machine perfusion of transplant organs may improve organ viability [37].

The organ perfusion system utilised Soltran clinical organ preservation solution (also known as hypertonic citrate solution), which may have influenced the cavity aspirate. This simple solution is based on high concentrations of citrate as the major anion [38], which is also an efficient chelator of calcium ions. The standard collagenase digestion systems are based on two-step perfusions in which an initial perfusion is made with calcium-free solutions [39], on the basis that this starts the process of weakening attachments between the liver cells and extracellular matrix components. Removal of the Ca^2+^ in the first step helps to disrupt desmosomes, whilst the addition of the Ca^2+^ in the second step is required for the optimum collagenases activity [40]. Although we did not use enzymatic digestion, the presence of calcium chelators (Soltran solution) in our experimental setting may have had a synergistic effect on the histotripsy sonication in tissue dissociation and the viability of the contained cells.

To our knowledge, our group is the first to describe the use of a perfused organ system for modelling the in vivo environment of liver histotripsy and the first to analyse the histotripsy cavity constituents. The perfusion system allows study of the perfusion fluid and temperature related to the type of liver [41] and histotripsy parameters with the intention of optimising hepatocyte yield and viability.

### 4.3. The Cell Isolates, Culture, and Cell Viability

Our study revealed that, following treatment with histotripsy, the core lesion contained live cells that when cultured not only retained their morphology and structure but also could divide. The initial isolate contained a mixture of cells and extracellular debris, the latter being removed following centrifugation.

The subsequent cell isolates contained hepatocytes and a small number of other cells such as sinusoidal endothelial cells, biliary canalicular cells and fibroblasts. These cells could have been separated from mature hepatocytes by low-speed centrifugal washes [2,3,4,5,6,7,9,42]. This was not done in the current experiments as we considered that mixed cell cultures might help sustain mature hepatocytes [43]. Of the cultured cells, 16% were alive on day 1 but remained alive beyond 7 days in standard culture conditions. Overall, the number of isolated liver cells almost doubled by day 7 and metabolic activity peaked a week post culture. After a week, the number of live cells started to decrease in number, most probably reflective of the lack of space in the plate. Mature hepatocytes could be identified in the cultures as typical large hexagonal cells, which by day 7 had aggregated in areas of the culture forming a natural 3D network.

It is known that primary hepatocytes in culture can lose their cuboidal morphology and liver specific function over a few days [44]. In our experimental culture, the hepatocytes retained morphology and function for at least 1 week. We considered that the success of culturing hepatocytes from the histotripsy aspirate might be related to the aspirate containing some extracellular matrix and some non-parenchymal cells (NPC). The co-culture of hepatocytes with NPCs has previously been shown to improve hepatocyte function significantly and may promote angiogenesis [45]. Additionally, hepatocytes and NPC in vitro co-cultivation can mimic the native hepatic microenvironment [46,47]. Additionally, in vitro co-cultivation of hepatocytes and nonparenchymal cells has been used to preserve and modulate the hepatocyte phenotype and function [48,49].

In our cell culture experiments, there might have been limited natural deposition of collagen by the mixed small cell components present along with hepatocytes, thus aiding the natural extra-cellular matrix (ECM) deposition and cell network formation. In contrast, the conventional hepatocyte isolation techniques discard non-parenchymal cells through post-enzymatic digestion washing steps. Our isolation and culturing method allowed both parenchymal and non-parenchymal cells to survive the process and in turn contribute towards the synthesis of ECM post culturing. Non-parenchymal cells such as sinusoidal endothelial cells and Kupffer cells are involved in secretion of bioactive factors and ECM components [50,51,52,53]. Kupffer cells not only enhance hepatocyte growth factor expression but can also produce growth factors, metalloproteinases, elastase, collagenase and fibronectin [53]. As the cell numbers in the current experiments were insufficient for flow cytometry, immunofluorescence staining will be used in future experiments to further quantify and characterise the cultured cells. Immunofluorescence would allow better identification of cell types isolated and provides a more accurate count of cells cultured than the current method used.

The CTG assay used in this study has been specifically designed to measure ATP levels in 3D cell cultures. This was because of the observation of cells clumping together. The fact that we observed cell clumping suggests that the viable hepatocytes were organising into 3D clusters. This structuring in vitro allows for interaction with the cell-ECM cross talk and may have an impact on the longevity of cell cultures. We and others have previously demonstrated that 3D hepatospheres retain morphology, live longer and have increased functionality compared with single-layer or 2D in vitro hepatocyte cultures [49,53,54,55].

The CTG assay measured mitochondrial activity and ATP production; hence the degree of luminescence was proportional to the number of live cells within the wells. The fact that these cells showed metabolic activity almost 21 days post histotripsy and culture suggests that cells isolated under these circumstances can remain alive beyond a week with their metabolic activity intact. This is an important finding in terms of hepatocyte preservation. Even though the culture medium was not optimised beyond 2 weeks, liver-derived cells were alive 21 days post culture. Longevity is important in cell transplantation especially when dealing with adult hepatocytes. Historically, methods such as supplementation of culture medium with non-physiological inducers (i.e., dimethyl sulfoxide or phenobarbital), as well as co-culture with other cell types, have been used to enhance longevity of hepatocytes in cell culture [56]. Additionally, the phalloidin/DAPI staining revealed intact cytoskeleton networks and clumped groups of cells at 7 days post culture. This may reflect connection of hepatocytes via intermediate filaments of cell cytoskeletons with their adjacent cell. Cells that have formed a network while maintaining their cytoskeleton may be more stable in maintaining their morphology in cell culture. Hepatocytes are rich in gap junctions, and cell proximity improves their interaction. In the liver, gap junctions are predominantly found in hepatocytes and play critical roles in virtually all phases of the hepatic life cycle, including cell growth, differentiation, liver-specific functionality and cell death [57,58].

## 5. Limitation of Current Study, Further Evaluation, and Possible Implications for Therapy

There are several aspects of the current study that may have affected the outcome of the histotripsy cavity analysis. Although the histotripsy treatment parameters had previously been studied [32] and the requirements of the current study had been modelled to produce a focal and superficial cavity, the US parameters had not been altered to assess the influence of histotripsy on cavity contents. Furthermore, the studies were done on porcine liver, which, whilst it bears many similarities to human liver, does have a different gross and microscopic anatomy [59]. This suggests that the experimental work should be repeated with human livers, possibly organs retrieved for the purpose of transplantation, but which are deemed unsuitable for clinical use. The cell viability studies were objective, but the cell characterisation was based on microscopic cell appearances analysis. Although this would be standard in clinical practice, confirmation of hepatocytes and other non-parenchymal cells could be further confirmed by characteristic synthetic functions and cell markers [58].

The finding of viable hepatocytes following histotripsy and their ability to survive, expand and replicate in subsequent culture is a unique finding that could have implications for research and clinical practice. Apart from cell transplantation, availability of viable human hepatocytes is essential for testing new systemic therapies to evaluate hepatotoxicity and may help accelerate scientific studies in other disciplines. The ability to produce a cavity in the liver adjacent to but not within the circulation is an ideal circumstance for implantation of transplant cells [57,58]. The finding of viable cells in the cavity aspirates raises the possibility of histotripsy being used as a method for cell isolation. Automation of the methods used in this study could be achieved along with optimisation of histotripsy parameters for this purpose.

Although this paper has its own limitations, it brings to light the possible application of histotripsy in cell isolation and transplantation. Future studies should aim to improve parameterisation of histotripsy and aim to improve culture techniques for cells isolated using histotripsy.

## Figures and Tables

**Figure 1 bioengineering-10-00276-f001:**
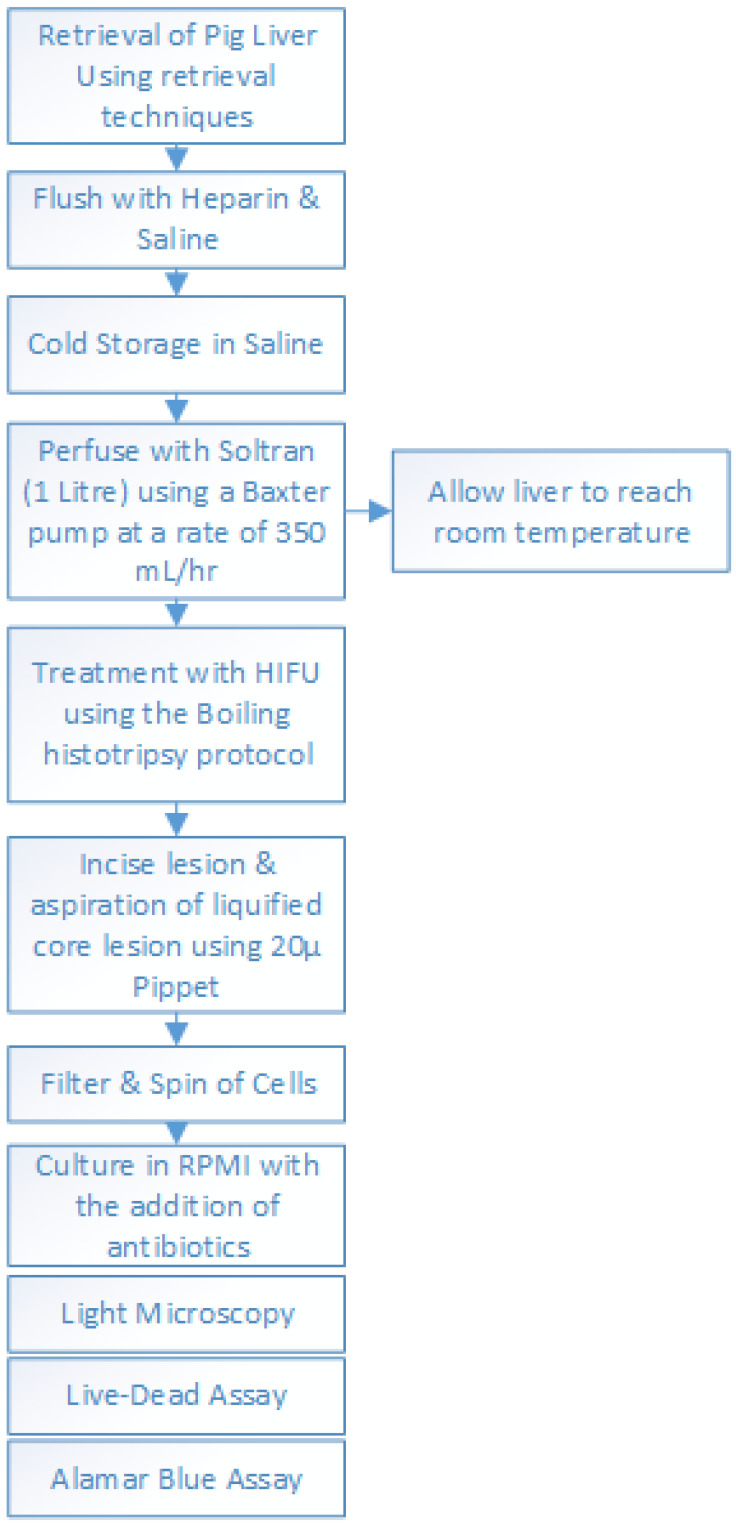
Experimental protocol for the hepatocyte cell isolation from perfused pig livers using HIFU device.

**Figure 2 bioengineering-10-00276-f002:**
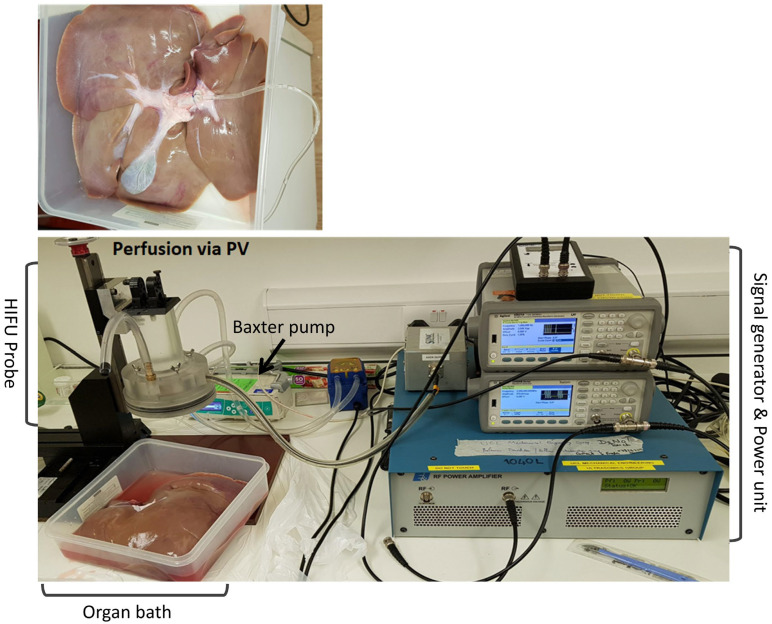
Experimental set up for HIFU probe and histotripsy protocol. Retrieved pig liver is placed in an organ bath and cannulated via portal vein for perfusion with Soltran solution with a Baxter pump. The perfusate is collected in the organ bath, which is then drained out of bath to ensure the probe is not submerged under accumulating perfusate fluid. The perfused liver is placed under the HIFU probe. The signal generator and power unit are adjusted to a set parameter for this experiment.

**Figure 3 bioengineering-10-00276-f003:**
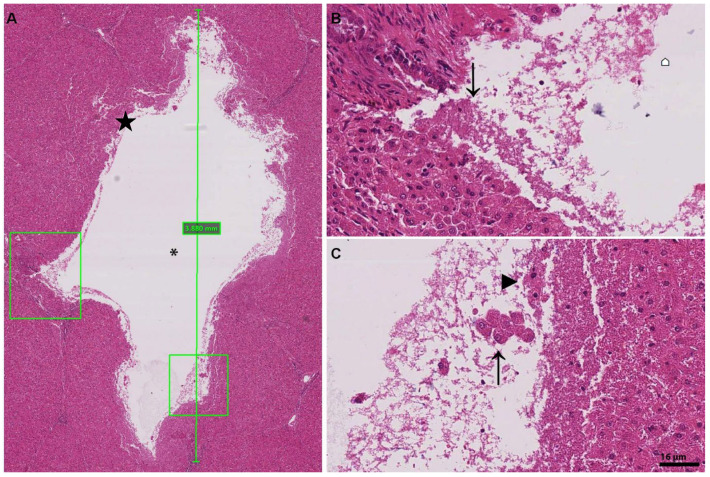
H&E cross-section staining of the HIFU histotripsy lesion. (**A**) Example of a lesion with its core suspension aspirated for analysis. The * marks the aspirated core of lesion where cell suspension would have been and HIFU creates a cavity of about 3.8 mm in size. The two square areas are zoomed in, the left box corresponding to (**B**) and the right box to (**C**), which show intact hepatocytes detached by the force of histotripsy. The force generated is focused and the adjacent cellular and extracellular components are preserved. (**B**) Reveals the impact of the histotripsy force, resulting in finely fragmented cells (**⌂**) and causing mechanical disturbance of adjacent cells. The arrow marks damage to a nearby vessel. (**C**) The force not only results in destruction of cellular and extracellular components into fine granules, but it can also result in the bursting of the cell membrane (**►**), whereas other cells escape (**↑**) the full impact of the force generated and are either in clusters of intact cells or as individual cells in the core suspension.

**Figure 4 bioengineering-10-00276-f004:**
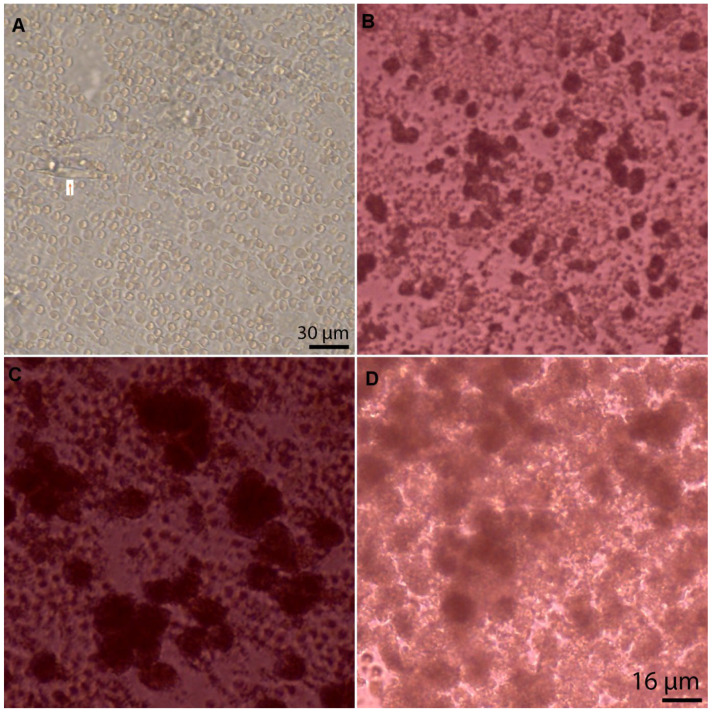
Harvest from each lesion demonstrating different quantity and type of cells being extracted. (**A**) Cell extract 24 h post culture prior to spin reveals a mixture of different cell types. Arrow points towards a fibroblast (20× zoom); (**B**) 2 days post culture; (**C**) 4 days post culture; (**D**) 6 days post culture (40× zoom).

**Figure 5 bioengineering-10-00276-f005:**
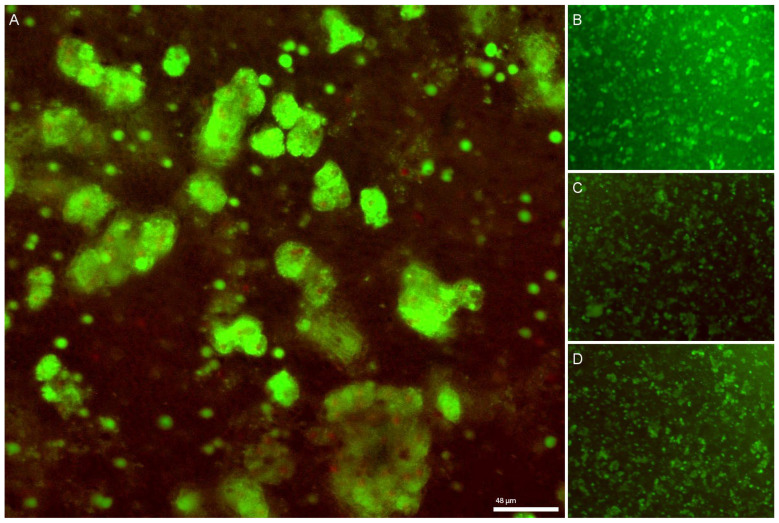
(**A**) Live-dead staining 24 h post culture reveals viable liver cells (green fluorescence) along with debris and dead cells (red stains) (20× zoom). (**B**–**D**) Cells in other wells demonstrating a mixture of live and dead cells.

**Figure 6 bioengineering-10-00276-f006:**
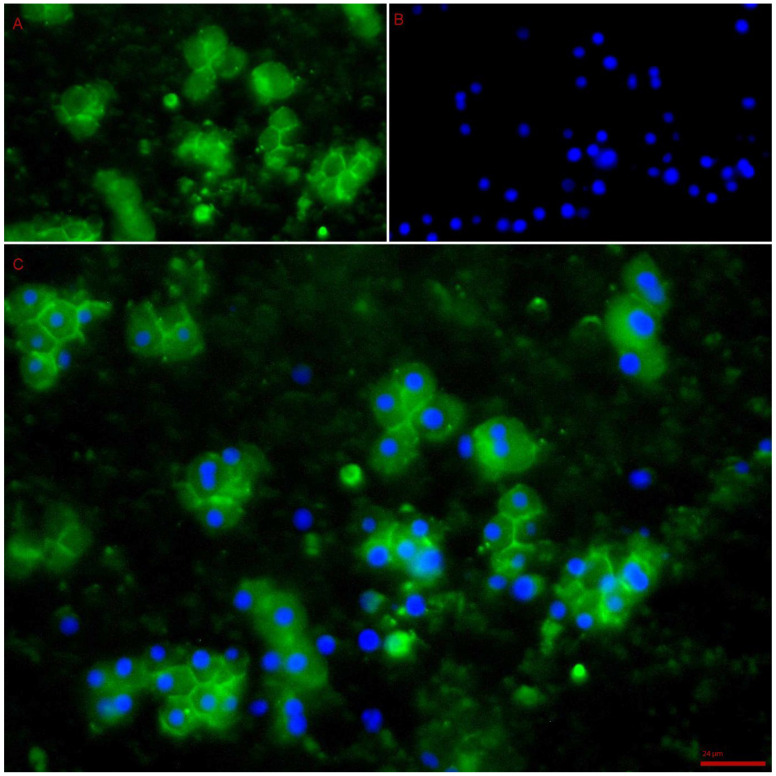
Phalloidin/DAPI stain 7 days post culture revealing clusters of live cells with preserved cytoskeleton. Blue: live cell nuclei; green: cytoskeleton of hepatocytes. (**A**) Green fluorescence showing the cytoskeleton of the hepatocytes, revealing intact cytoskeleton network and clumped group of cells at 7 days post culture. (**B**) Blue fluorescence revealing nuclei of live hepatocytes. (**C**) Combined fluorescence of the superimposed nuclei and cytoskeletal staining, revealing live cells clumped together at 7 days post culture while preserving their skeletal integrity. The cells seem to conform to a 3D structure and are bound to a natural matrix evident on the phalloidin/DAPI staining. (40× Zoom EVOS microscopy).

**Figure 7 bioengineering-10-00276-f007:**
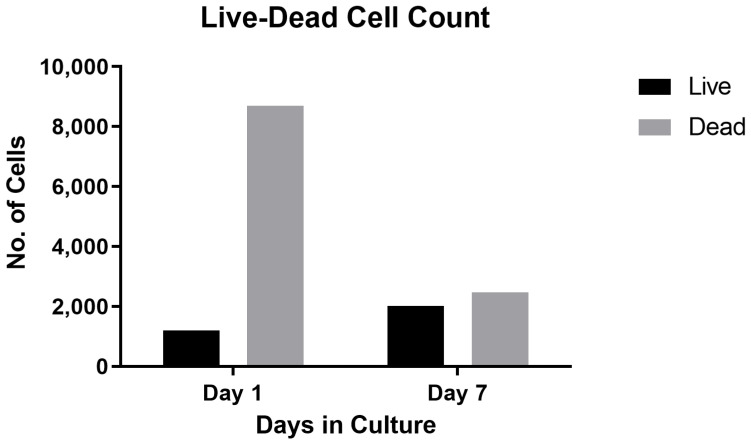
Live-dead cell count analysis. Using ImageJ software (live-dead plugin), number of liver cells were quantified on day 1 and day 7. There is an increase in the number of live cells 7 days post culture; almost double the number of live cells observed a week after culture, and this indicates cell viability and replication.

**Figure 8 bioengineering-10-00276-f008:**
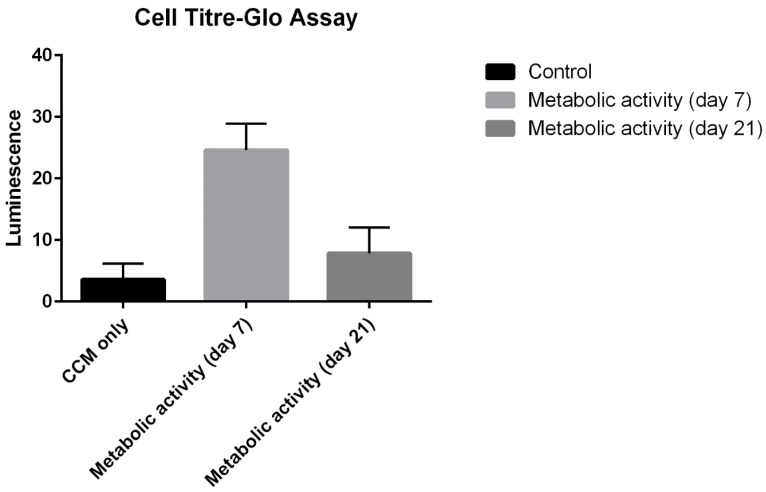
Cell Titre-Glo Metabolic assay. The cultured cells had a high metabolic activity a week post culture and this reduced by 21 days post culture. Even at 21 days, the cultured cells demonstrated metabolic activity.

**Table 1 bioengineering-10-00276-t001:** Analysis of live and dead cells at day 1 and day 7 post culture.

	Live	Dead	Total	% Live
**Day 1**	1206	8690	9896	12.2
**Day 7**	2022	2460	4482	45.1

## Data Availability

Not applicable.

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
