# Peer review of "Liver Ultrasound Histotripsy: Novel Analysis of the Histotripsy Site Cell Constituents with Implications for Histotripsy Application in Cell Transplantation and Cancer Therapy"

_bioengineering, 2023, doi:10.3390/bioengineering10020276_

Round 1

Reviewer 1 Report

Section 2.1.1 Report on the approximate size of the focal zone, and the peak negative/positive pressures, and pulse duration

2.1.3 What was the depth of the ablation zone? How much tissue did the focused ultrasound pulse have to transverse?

Section 3.1: Why 29 C instead of physiologic temperature of 37 C?

Results: General. A primary limitation of this study is the lack of bubble cloud monitoring, either via B-mode ultrasound imaging or acoustic emissions. These findings should be clarified in the context of the lack of bubble detection methods, meaning an insufficient amount of bubble activity may have been applied to ensure complete fractionation of the focal zone.

Line 312: Why a depth of 0.5 cm? Is there some need to target at this depth (or deeper) based on the application?

Line 411: Reporting that histotripsy may cause an increase in metastatic spread of the disease is speculative in this study, and inconsistent with the literature. The authors should also consider the context of their results relative to data collected with shock scattering histotripsy regarding vessel perforation:

Vlaisavljevich E, Owens G, Lundt J, Teofilovic D, Ives K, Duryea A, Bertolina J, Welling T H and Xu Z 2017 Non-Invasive Liver Ablation Using Histotripsy: Preclinical Safety Study in an In~Vivo Porcine Model Ultrasound Med. Biol. 43 1237–51

Authors should also consider that the results of vessel perforation may be specific to boiling histotripsy, which results in large temperature changes that will make the vessel more compliant and susceptible to perforation. Shock scattering or intrinsic threshold histotripsy, in contrast, causes no changes in tissue temperature and therefore reduces the likelihood of vasacular damage.

Line 420: Ruger et al also noted viable cells within the ablation zone, particularly near vascular-rich regions targets:

Ruger L N, Hay A N, Gannon J M, Sheppard H O, Coutermarsh-Ott S L, Daniel G B, Kierski K R, Ciepluch B J, Vlaisavljevich E and Tuohy J L 2022 Histotripsy Ablation of Spontaneously Occurring Canine Bone Tumors In Vivo IEEE Trans. Biomed. Eng. 1–12 Online: https://ieeexplore.ieee.org/document/9830057/

Section 4.2: Histotripsy damage is isothermal in nature (see figure 8 in Bader K B, Vlaisavljevich E and Maxwell A D 2019 For Whom the Bubble Grows: Physical Principles of Bubble Nucleation and Dynamics in Histotripsy Ultrasound Therapy Ultrasound Med. Biol. 45 1056–80), so this discussion of heat-sink is nonsensical

Section 4.3: See comment on bubble detection, and recent study by Ruger

Author Response

Response 1:

Section 2.1.1 Report on the approximate size of the focal zone, and the peak negative/positive pressures, and pulse duration

The approximate Size of the focal zone and the pulse duration are now given in this section. The peak positive/negative pressures are given in section 2.1.2.

Response 2:

2.1.3 What was the depth of the ablation zone? How much tissue did the focused ultrasound pulse have to transverse?

The depth of the ablation zone is now given in section 2.1.1.

Response 3:

Section 3.1: Why 29 C instead of physiologic temperature of 37 C?

Although we did not actively re-warm the organ on arrival with the aid of warm perfusion solution. The Temperature was allowed to reach a range close to physiological parameters. One rationale for not allowing the organ to reach the physiological temperature range was to prevent or reduce the enzyme activity that would ultimately result in tissue breakdown through autolysis or change in cell activity. We wished to slowly re-warm the liver from hypothermia to room temperature to minimise tissue injury but did not attempt reproduce normal physiological body temperature to minimise this. Sub-normothermic reperfusion is an effective method of minimising organ injury.

Machine perfusion preservation versus static cold storage for deceased donor kidney transplantation.Tingle SJ, Figueiredo RS, Moir JA, Goodfellow M, Talbot D, Wilson CH.Cochrane Database Syst Rev. 2019 Mar 15;3(3):CD011671. doi: 10.1002/14651858.CD011671.pub2.PMID: 30875082

We did not constantly monitor the temperature, but experiments were carried out once the temperature had reached roughly room temp which was within a range of 24-30C. Typically the room temperature where we carried out the experiments was around 29C.

It is well known that Temp can affect the outcome of the cell culture. Mammalian cells show a remarkably high viability and a decreased proliferation rate at temperatures between 27 °C and 32 °C.

Kaufmann H, Mazur X, Fussenegger M, Bailey JE. Influence of low temperature on productivity, proteome and protein phosphorylation of CHO cells. Biotechnol Bioeng. 1999 Jun 5;63(5):573-82. doi: 10.1002/(sici)1097-0290(19990605)63:5<573::aid-bit7>3.0.co;2-y. PMID: 10397813.

Giard DJ, Fleischaker RJ, Fabricant M. Effect of temperature on the production of human fibroblast interferon. Proc Soc Exp Biol Med. 1982;170:155–159

Cultured mammalian cells can tolerate temperatures of 4C and can tolerate freezing temperatures (-196C) for preservation with the appropriate protocol. However, they cannot survive in temperatures 2 degrees above average physiological temperature.

Oguchi, S., Saito, H., Tsukahara, M., Tsumura, H. (2003). Control of Temperature and pH Enhances Human Monoclonal Antibody Production in CHO Cell Culture. In: Yagasaki, K., Miura, Y., Hatori, M., Nomura, Y. (eds) Animal Cell Technology: Basic & Applied Aspects. Animal Cell Technology: Basic & Applied Aspects, vol 13. Springer, Dordrecht. https://doi.org/10.1007/978-94-017-0726-8_29

Effects of temperature on growth rate of cultured mammalian cells (L5178Y); I Watanabe, S Okada - The Journal of cell biology, 1967 - rupress.org

Response 4:

Results: General. A primary limitation of this study is the lack of bubble cloud monitoring, either via B-mode ultrasound imaging or acoustic emissions. These findings should be clarified in the context of the lack of bubble detection methods, meaning an insufficient amount of bubble activity may have been applied to ensure complete fractionation of the focal zone.

In our previous work, reported in reference [16], we have monitored acoustic emissions during boiling histotripsy using a PCD. The PCD results reported there show significant broadband emissions due to inertial cavitation activity after the formation of the bubble cloud. In fact, the point in time at which both a boiling bubble is created and the cavitation cloud is excited can be seen in the analysis of PCD signal spectrograms. In view of this, we have added a sentence to section 4.1 to clarify that in view of our previous work we expected to have excited sufficient bubble activity for complete tissue fractionation.  

Response 5:

Line 312: Why a depth of 0.5 cm? Is there some need to target at this depth (or deeper) based on the application?

There is no specific medical need to target at this depth. This is based on the characteristics of the transducer + cone used, that give us a depth of 5.6 mm for the centre of the lesion below the surface with the cone in contact with the tissue sample.

Response 6:

Line 411: Reporting that histotripsy may cause an increase in metastatic spread of the disease is speculative in this study, and inconsistent with the literature. The authors should also consider the context of their results relative to data collected with shock scattering histotripsy regarding vessel perforation:

 Vlaisavljevich E, Owens G, Lundt J, Teofilovic D, Ives K, Duryea A, Bertolina J, Welling T H and Xu Z 2017 Non-Invasive Liver Ablation Using Histotripsy: Preclinical Safety Study in an In~Vivo Porcine Model Ultrasound Med. Biol. 43 1237–51

We agree that the question of metastatic spread is speculative at this stage, as our experiments were not designed to show this. However, prior work has raised concerns about whether cavitation and mechanical forces that occur with histotripsy might result in unwanted metastatic dissemination of the tumour. A recent study on rodent models highlights the insufficient evidence regarding the effects of histotripsy on the risk of recurrence and metastases following tumour debulking (Cancers. 14(7):1612). Whilst this study suggests that histotripsy may not increase the risk of developing metastases after treatment, the authors acknowledge that further work is required to ascertain this. Furthermore, it is a possibility that the presence of an enhanced immune response and the abscopal effect can reduce the importance of any cancer cell metastasis.

A sentence acknowledging the work of Vlaisavljevich et al. on damage to vascular walls using intrinsic threshold histotripsy has been added to section 4.1.

Response 7:

Authors should also consider that the results of vessel perforation may be specific to boiling histotripsy, which results in large temperature changes that will make the vessel more compliant and susceptible to perforation. Shock scattering or intrinsic threshold histotripsy, in contrast, causes no changes in tissue temperature and therefore reduces the likelihood of vascular damage.

The evidence presented by Vlaisavljevich et al (Ultrasound Med. Biol. 43 1237–51) suggest otherwise and we quote: ‘’Localized perforation of the vein wall was observed in some subjects (heparinized: 7 of 11 subjects, non-heparinized: 5 of 11 subjects), with the regions of microscopic vessel rupture occurring adjacent to the histotripsy lesions. The size of the vessel ruptures ranged from ∼50 to ∼500 μm’’.

Response 8:

Line 420: Ruger et al also noted viable cells within the ablation zone, particularly near vascular-rich regions targets:

Ruger L N, Hay A N, Gannon J M, Sheppard H O, Coutermarsh-Ott S L, Daniel G B, Kierski K R, Ciepluch B J, Vlaisavljevich E and Tuohy J L 2022 Histotripsy Ablation of Spontaneously Occurring Canine Bone Tumors In Vivo IEEE Trans. Biomed. Eng. 1–12 Online: https://ieeexplore.ieee.org/document/9830057/

Thank you for this reference. Although the application is very different in terms of the target tissue properties, this would be a useful reference to include. We have done this is section 4.1.

Response 9:

Section 4.2: Histotripsy damage is isothermal in nature (see figure 8 in Bader K B, Vlaisavljevich E and Maxwell A D 2019 For Whom the Bubble Grows: Physical Principles of Bubble Nucleation and Dynamics in Histotripsy Ultrasound Therapy Ultrasound Med. Biol. 45 1056–80), so this discussion of heat-sink is nonsensical

We intended to discuss and highlight that the heat sink effect should not affect boiloing histotripsy. We have revised and shortened this section as suggested. Although we believe we have not said anything different here, as it has been pointed out in this section that due to the difference in timescales, perfusion is not expected to affect boiling histotripsy. We have removed two sentences from section 4.2 to avoid any confusions on this point.   

 Response 10:

Section 4.3: See comment on bubble detection, and recent study by Ruger

Thank you for bringing the Ruger et al paper to our attention. 

Reviewer 2 Report

Please, find reviewer comment and annotation files.

Title:

Liver ultrasound histotripsy: Novel analysis of the histotripsy site cell constituents with

mplications for histotripsy application in cell transplantation and cancer therapy.

The major contribution is to use “ultrasound histotripsy to provide a nidus in the liver for

direct cell transplantation offers a new approach to overcoming key limitations to current cell

therapy”. It is an outstanding contribution and the methods and results are sounding.

Observations such as (line 416) “Our study which has revealed for the first time the

presence of live cells within the core lesion post histotripsy could explain the finding of

tumour regrowth if histotripsy for cancers similarly resulted in a liquid core of viable cancer

cells.” deserves further investigation by the community.

Only minor changes should be considered as follow.

Line 37 – There is an extra “)”

The fluid from each lesion was aspirated and cultured in medium (RPMI) at 370 in an incubator)

line 100

In the sentence below, I suggest using commas instead of [ ].

“been based on the collagenase digestion technique [originally developed by Berry & Friend(20)] ,”

line 110

The sentence seems incomplete. Please, revise it.

“To investigate the possible use of histotripsy as a novel technology in extracting cells and

assessing the lesions histologically for their suitability as a nidus for cell transplantation. ”

Line 123

“using the protocol described (Figure 1)”. Not everything in the Figure 1 was described

in the text. Please, do so. While doing it, refer to the Figure.

Figure 1

I suggest aggregating BOXES which refer to a single action. Example: Perfuse with 1L

Soltran solution using Baxter pump at 350 mL/h. By the way, pay attention to the units L

(liter) and h(hour).

line 140

The use of term “square wave”

Although the term square wave is generally used in two state waveforms, strictly speaking,

a square wave is one that has duty cycle equal to 50%. On the other hand, a rectangular

waveform is a periodic two state waveform that has duty cycle different of 50%, as is the

case for the 1% duty cycle wave used.’

line 139

In the sentence :

“The first function generator is set to generate 50 cycles of a 1 Hz square wave with 1% duty

cycle. This triggers the second function generator, that outputs a 2 MHz sinusoidal wave into

the RF power amplifier.”

I assume the 1% duty cycle rectangular wave (FROM PAPER LINE 140: 1 Hz square wave

with 1% duty cycle) has such a duty cycle (i.e., 10 ms ON, 990 ms OFF) only because the

pulse is used for triggering the sinusoidal wave. Therefore, in 10 ms it is possible to have

20,000 sinusoidal periods.

It is not clear to this reviewer what the authors mean by “measuring a power of 150 W”.

Reading the Sonic Concepts website, it looks like what is given is the “Average power”.

From the site

https://openstax.org/books/university-physics-volume-2/pages/15-4-power-inan-

ac-circuit allows to make sure what is being measured if you do have all

the information. Please, review whether the measurement was 150W. The

site

https://www.rfglobalnet.com/doc/basics-of-power-measurement-average-or-p

eak-0001 gives an idea of what I am talking about. In this paper, authors have

used sine wave burst instead of a modulated pulse as shown the site.

I understand that what really goes to the transducer are BURSTS of 20,000 cycles of a

2MHz SINE WAVE. Actually, 50 bursts, 1 second apart from each other.

I understand this is important since the acoustic simulation cited in line 154 relies upon the

electrical power given to provide the Pressure estimation given in line 161. Also, I

understand a simulator ESTIMATES a value, thus my suggestion to change the word in line

161.

Please, revise the use of “space” between the numeral and the unit. Example, in line 155 it is

used 150W (no space) while in the same line it is used 2 MHz (with space)

114-123 and 164-173

Lines 114-123 are somewhat repetitive with the lines 164-173 (see yellow highlighted lines in

pdf file). Also, some textual explanation that was lacking for Figure 1 is present in this

second comment on the same issue. I suggest reorganizing the information, in order to use

a single explanation and reference to Figure 1.

line 179

Revise mL/h versus ml/hr units

line 301

hour unit must be corrected - at 350mL/hr

Please, review the RESULTS section. Part of the text does not represent RESULTS but

METHODS instead.

Example line 300-307:

Livers were perfused with non-oxygenated perfusion fluids at 350mL/hr via the portal vein

and the median starting temperature of the perfusate was 29℃. The experiments were

repeated with 5 different porcine livers and similar histotripsy parameters of the HIFU

machine were used. We analysed 130 individual lesions from these 5 livers using this

protocol. During perfusion all livers maintained a degree of bile production suggesting they

were viable livers capable of active bile production and excretion. Following treatment with

histotripsy each lesion was bisected and the fluid contents at the core of lesion was then

aspirated. Time taken to aspirate each lesion was less than a minute. To increase the

volume of cell suspension the aspirate from three sequential lesion was combined in each

well of the 96 well culture plate.

On the other hand, in the section “3.2 Histology of lesions”, most of it represent RESULTS

since it is describing to the reader the result of the use of HIFU-based technique. Therefore,

it is only necessary to verify what is really RESULT and what part can be moved to a

subsection of section 2 (Materials and Methods)

Metabolic activity of the cultured cells analysed using cell titre-Glo assay showed an activity

peak 7 days post culture (Figure 8) and further confirm survival of cells. This was a

significant increase when compared to control luminescence (medium alone) (P<0.0001);

Luminescence: 3.57 RLU2).

line 366. I suggest using the AVERAGE number of … if this is the case.

The ??? number of live cells per well almost doubled between baseline and day 7(Day 1:

1206 cells to Day 7: 2022 per well)

line 370. Consider re-writing this sentence.

Metabolic activity of the cultured cells analysed using cell titre-Glo assay showed an activity

peak 7 days post culture (Figure 8) and further confirm survival of cells.

In methods, only 3 points in the timeline were sampled for metabolic activity. It does not

seem possible to say that 7th day was the one with the higher activity. If a test were made

with measurements with time intervals of 1 day, it might be possible. However, it is possible

to say that, considering the samples taken in time, the 7th day sample presented the higher

activity.

Line 373 - there is an extra “2)”

line 412. the sentence is lacking a verb

Break in vasculature in the clinical setting could ???? of concern in terms of bleeding if

being used for focal cavity …

The section Treatment dose and the nature of the liver lesion based on histology

is an important section. On the other hand, it discusses observations that might not be

reported. For example, HOW/WHY 1% duty cycle was chosen should be presented in

materials and methods to be discussed here.

In general, the DISCUSSION section has amazing scientific considerations. Just make sure

it DISCUSS something that was previously presented in the paper.

line 566

It is lacking a REFERENCE NUMBER

Author Response

Response To comments:

Title:

Liver ultrasound histotripsy: Novel analysis of the histotripsy site cell constituents with mplications for histotripsy application in cell transplantation and cancer therapy.

The major contribution is to use “ultrasound histotripsy to provide a nidus in the liver for direct cell transplantation offers a new approach to overcoming key limitations to current cell therapy”. It is an outstanding contribution and the methods and results are sounding.

Observations such as (line 416) “Our study which has revealed for the first time the presence of live cells within the core lesion post histotripsy could explain the finding of tumour regrowth if histotripsy for cancers similarly resulted in a liquid core of viable cancer cells.” deserves further investigation by the community.

Thank you for the encouraging comments.

Response 1:

Only minor changes should be considered as follow.

Line 37 – There is an extra “)”

The fluid from each lesion was aspirated and cultured in medium (RPMI) at 370 in an incubator)

This has been amended.

Response 2:

line 100

In the sentence below, I suggest using commas instead of [ ].

“been based on the collagenase digestion technique [originally developed by Berry & Friend(20)] ,”

Amended accordingly.

Response 3:

line 110

The sentence seems incomplete. Please, revise it.

“To investigate the possible use of histotripsy as a novel technology in extracting cells and

assessing the lesions histologically for their suitability as a nidus for cell transplantation. ”

The sentence has been revised to read:

“To investigate the possible use of histotripsy as a novel technology in extracting cells and assessing the lesions histologically for their suitability as a nidus for cell transplantation we carried out an ex-vivo perfused pig liver experiment.”

Response 4:

Line 123

“using the protocol described (Figure 1)”. Not everything in the Figure 1 was described

in the text. Please, do so. While doing it, refer to the Figure.

This has been corrected. The citing for figure 1 has been moved to the second paragraph to include what has been described in the diagram. We have added an extra sentence to encompass the subsequent stages of the protocol. We have avoided a detailed description, so we don’t repeat the methodology for each section.

Response 5:

Figure 1

I suggest aggregating BOXES which refer to a single action. Example: Perfuse with 1L

Soltran solution using Baxter pump at 350 mL/h. By the way, pay attention to the units L

(liter) and h(hour).

The diagram for protocol has been revised as suggested to one box. A revised version has been added to the draft instead of the old diagram. Units have been doubled checked for correctness.

Response 6:

line 140

The use of term “square wave”

Although the term square wave is generally used in two state waveforms, strictly speaking,

a square wave is one that has duty cycle equal to 50%. On the other hand, a rectangular

waveform is a periodic two state waveform that has duty cycle different of 50%, as is the

case for the 1% duty cycle wave used.’

We agree and have corrected this to a rectangular waveform.

Response 7:

line 139

In the sentence :

“The first function generator is set to generate 50 cycles of a 1 Hz square wave with 1% duty

cycle. This triggers the second function generator, that outputs a 2 MHz sinusoidal wave into

the RF power amplifier.”

I assume the 1% duty cycle rectangular wave (FROM PAPER LINE 140: 1 Hz square wave

with 1% duty cycle) has such a duty cycle (i.e., 10 ms ON, 990 ms OFF) only because the

pulse is used for triggering the sinusoidal wave. Therefore, in 10 ms it is possible to have

20,000 sinusoidal periods.

The reviewer’s understanding is correct. We have re-written this section to make it read better and add clarity.

It is not clear to this reviewer what the authors mean by “measuring a power of 150 W”.

Reading the Sonic Concepts website, it looks like what is given is the “Average power”.

From the site

https://openstax.org/books/university-physics-volume-2/pages/15-4-power-inan-

ac-circuit allows to make sure what is being measured if you do have all

the information. Please, review whether the measurement was 150W. The

site

https://www.rfglobalnet.com/doc/basics-of-power-measurement-average-or-p

eak-0001 gives an idea of what I am talking about. In this paper, authors have

used sine wave burst instead of a modulated pulse as shown the site.

I understand that what really goes to the transducer are BURSTS of 20,000 cycles of a

2MHz SINE WAVE. Actually, 50 bursts, 1 second apart from each other.

I understand this is important since the acoustic simulation cited in line 154 relies upon the

electrical power given to provide the Pressure estimation given in line 161. Also, I

understand a simulator ESTIMATES a value, thus my suggestion to change the word in line

161.

Again, the reviewer’s understanding is correct, but we acknowledge that this section should have been more clearly written As such, we have modified it to clarify points such as what we mean by peak power versus average power. We have also changed the word ‘predict’ to ‘estimate’.

Response 8:

Please, revise the use of “space” between the numeral and the unit. Example, in line 155 it is

used 150W (no space) while in the same line it is used 2 MHz (with space)

114-123 and 164-173

Amended accordingly.

Response 9:

Lines 114-123 are somewhat repetitive with the lines 164-173 (see yellow highlighted lines in

pdf file). Also, some textual explanation that was lacking for Figure 1 is present in this

second comment on the same issue. I suggest reorganizing the information, in order to use

a single explanation and reference to Figure 1.

Amended as per suggestion

Response 10:

line 179

Revise mL/h versus ml/hr units

Corrected

Response 11:

line 301

hour unit must be corrected - at 350mL/hr

Corrected

Response 12:

Please, review the RESULTS section. Part of the text does not represent RESULTS but

METHODS instead.

Example line 300-307:

Livers were perfused with non-oxygenated perfusion fluids at 350mL/hr via the portal vein

and the median starting temperature of the perfusate was 29℃. The experiments were

repeated with 5 different porcine livers and similar histotripsy parameters of the HIFU

machine were used. We analysed 130 individual lesions from these 5 livers using this

protocol. During perfusion all livers maintained a degree of bile production suggesting they

were viable livers capable of active bile production and excretion. Following treatment with

histotripsy each lesion was bisected and the fluid contents at the core of lesion was then

aspirated. Time taken to aspirate each lesion was less than a minute. To increase the

volume of cell suspension the aspirate from three sequential lesion was combined in each

well of the 96 well culture plate.

Amended and relevant sentences have been moved to methods. The revision now reads more of results description.

On the other hand, in the section “3.2 Histology of lesions”, most of it represent RESULTS

since it is describing to the reader the result of the use of HIFU-based technique. Therefore,

it is only necessary to verify what is really RESULT and what part can be moved to a

subsection of section 2 (Materials and Methods)

This is in keeping with the results produced because of histotripsy sonication. The methods for this are described in the methods section, and there is no overlap/repetition.

Response 11:

Metabolic activity of the cultured cells analysed using cell titre-Glo assay showed an activity

peak 7 days post culture (Figure 8) and further confirm survival of cells. This was a

significant increase when compared to control luminescence (medium alone) (P<0.0001);

Luminescence: 3.57 RLU2).

line 366. I suggest using the AVERAGE number of … if this is the case.

The ??? number of live cells per well almost doubled between baseline and day 7(Day 1:

1206 cells to Day 7: 2022 per well)

line 370. Consider re-writing this sentence.

Addressed, the word average has been added.

Response 12:

Metabolic activity of the cultured cells analysed using cell titre-Glo assay showed an activity

peak 7 days post culture (Figure 8) and further confirm survival of cells.

In methods, only 3 points in the timeline were sampled for metabolic activity. It does not

seem possible to say that 7th day was the one with the higher activity. If a test were made

with measurements with time intervals of 1 day, it might be possible. However, it is possible

to say that, considering the samples taken in time, the 7th day sample presented the higher

activity.

Response 13:

Line 373 - there is an extra “2)”

Seems correct numbers cited

Response 14:

line 412. the sentence is lacking a verb

Break in vasculature in the clinical setting could ???? of concern in terms of bleeding if

being used for focal cavity …

Corrected

Response 15:

The section Treatment dose and the nature of the liver lesion based on histology

is an important section. On the other hand, it discusses observations that might not be

reported. For example, HOW/WHY 1% duty cycle was chosen should be presented in

materials and methods to be discussed here.

In general, the DISCUSSION section has amazing scientific considerations. Just make sure

it DISCUSS something that was previously presented in the paper.

This has now been addressed in discussion with the addition of the new lines 470 to 476.

Response 16:

line 566

It is lacking a REFERENCE NUMBER

Ref 29:

Khokhlova, T.D., Schade, G.R., Wang, YN. et al. Pilot in vivo studies on transcutaneous boiling histotripsy in porcine liver and kidney. Sci Rep 9, 20176 (2019). https://doi.org/10.1038/s41598-019-56658-7

Above is the citation for this part, which has been added.